# Whole-Genome Sequencing of a Colistin-Resistant *Acinetobacter baumannii* Strain Isolated at a Tertiary Health Facility in Pretoria, South Africa

**DOI:** 10.3390/antibiotics11050594

**Published:** 2022-04-28

**Authors:** Noel-David Nogbou, Mbudzeni Ramashia, Granny Marumo Nkawane, Mushal Allam, Chikwelu Lawrence Obi, Andrew Munyalo Musyoki

**Affiliations:** 1Microbiological Pathology Department, School of Medicine, Sefako Makgatho Health Sciences University, Pretoria 0204, South Africa; davnogbou2@gmail.com (N.-D.N.); ramashiambudzeni@gmail.com (M.R.); nkawanegranny@gmail.com (G.M.N.); 2Department of Genetics and Genomics, College of Medicine and Health Sciences, United Arab Emirates University, Al Ain 15551, United Arab Emirates; mushalallam@gmail.com; 3School of Sciences and Technology, Sefako Makgatho Health Sciences University, Pretoria 0204, South Africa; lawrence.obi@smu.ac.za

**Keywords:** *Acinetobacter baumannii*, colistin resistance, resistance mechanism, virulence factors, South Africa

## Abstract

Background: *Acinetobacter baumannii*’s (*A. baumannii*) growing resistance to all available antibiotics is of concern. The study describes a colistin-resistant *A. baumannii* isolated at a clinical facility from a tracheal aspirate sample. Furthermore, it determines the isolates’ niche establishment ability within the tertiary health facility. Methods: An antimicrobial susceptibility test, conventional PCR, quantitative real-time PCR, phenotypic evaluation of the efflux pump, and whole-genome sequencing and analysis were performed on the isolate. Results: The antimicrobial susceptibility pattern revealed a resistance to piperacillin/tazobactam, ceftazidime, cefepime, cefotaxime/ceftriaxone, imipenem, meropenem, gentamycin, ciprofloxacin, trimethoprim/sulfamethoxazole, tigecycline, and colistin. A broth microdilution test confirmed the colistin resistance. Conventional PCR and quantitative real-time PCR investigations revealed the presence of *adeB*, *adeR*, and *adeS*, while *mcr-1* was not detected. A MIC of 0.38 µg/mL and 0.25 µg/mL was recorded before and after exposure to an AdeABC efflux pump inhibitor. The whole-genome sequence analysis of antimicrobial resistance-associated genes detected beta-lactam: *bla_OXA-66_*; *_blaOXA-23_*; *bla_ADC-25_*; *bla_ADC-73_*; *bla_A1_*; *bla_A2_*, and *bla_MBL_*; aminoglycoside: *aph(6)-Id*; *aph(3″)-Ib*; *ant(3″)-IIa* and *armA)* and a colistin resistance-associated gene *lpsB*. The whole-genome sequence virulence analysis revealed a biofilm formation system and cell–cell adhesion-associated genes: *bap*, *bfmR*, *bfmS*, *csuA*, *csuA/B*, *csuB*, *csuC*, *csuD*, *csuE*, *pgaA*, *pgaB*, *pgaC*, and *pgaD*; and quorum sensing-associated genes: *abaI* and *abaR* and iron acquisition system associated genes: *barA*, *barB*, *basA*, *basB*, *basC*, *basD*, *basF*, *basG*, *basH*, *basI*, *basJ*, *bauA*, *bauB*, *bauC*, *bauD*, *bauE*, *bauF*, and *entE*. A sequence type classification based on the Pasteur scheme revealed that the isolate belongs to sequence type ST2. Conclusions: The mosaic of the virulence factors coupled with the resistance-associated genes and the phenotypic resistance profile highlights the risk that this strain is at this South African tertiary health facility.

## 1. Background

*Acinetobacter baumannii* (*A. baumannii*) is an opportunistic nosocomial Gram- negative nonmotile organism [1]. The bacteria can cause serious healthcare-associated infections, such as bacteremia, ventilator-associated pneumonia, urinary tract infection, meningitis, and skin and soft tissue infections associated with high mortality, mainly among intensive care unit hospitalized patients [2,3]. *A. baumannii* has emerged as an important clinical pathogen due to its ability to acquire and spread resistance-associated genes [4,5]. Globally, there is an increased report of colistin-resistant *A. baumannii* [6]. In South Africa, this raises public health concerns regarding the treatment of *A. baumannii* infections [7,8]. The precise mechanism of the action of colistin is not completely understood [9]. However, genomic investigations revealed that colistin induces rapid, complex perturbations of multiple key metabolic pathways in *A. baumannii*, leading to the disruption of the bacteria cell membrane [10,11,12]. Various molecular interactions, including mutation, structural modification, and enzyme overexpression that are chromosomally mediated [7,13], result in *A. baumannii’s* increased tolerance to colistin [14,15]. A mutation in the *pmrB/pmrA/pmrC* operon leads to a structural modification of the lipid A component of the LPS, which induces an increased tolerance to colistin [16]. Another described mutation in *lpxA, lpxC,* and *lpxD* genes encoding lipid A biosynthesis results in resistance to colistin due to the complete loss of LPS [17]. The presence of the insertion sequence, *ISAba11*, in *lpxA* or *lpxC* leads to the inactivation of LPS production, which results in decreased susceptibility to colistin [14]. Lastly, the overexpression of the phosphoethanolamine transferase enzyme drives the integration of the insertion element, *ISAbaI*, upstream of a *pmrC* homolog, *eptA*, which leads to colistin resistance [15]. Furthermore, plasmid-borne *mcr-1^_^9* genes confer resistance to colistin [7]; however, these genes have not yet been described in *A. baumannii* [18] In South Africa, the *mcr-1* variant gene has been reported in *Klebsiella pneumoniae* and *Escherichia coli* [19,20]. These bacteria are highly prevalent in hospital environments [21,22], subsequently offering a source for *mcr* gene uptake in the vicinity of *A. baumannii* [23,24]. Finally, the use of an active overexpressed adverse effect ATP-binding cassette (adeABC) efflux pump confers indiscriminate resistance to a wider class of antibiotics [25,26,27], including polymyxins such as colistin. The overexpression of the adeABC efflux pump can be triggered by genetic mutation occurring in *AdeR*, the regulatory gene, and *AdeS*, the sensor histidine kinase gene [28,29]. These two genes form the two-component system control of the adeABC efflux pump [28,29]. The system avoids the accumulation of drugs at the targeted site within the cells, leading to decreased susceptibility to antibiotics [24,30].

The success of *A. baumannii* as a nosocomial pathogen is also attributed to fundamental virulence mechanisms due to Acinetobacter chlorhexidine (AceI) efflux, RecA, and *A. baumannii* biofilm-associated proteins’ (Bap_Ab_) production [4,31,32]. Chlorhexidine is used as an antiseptic or disinfectant in hospitals to disrupt cell membranes and is active against Gram-positive and Gram-negative bacteria [33]. However, *A. baumannii* has been shown to actively pump chlorhexidine out of the cell using the AceI efflux protein [31], resulting in resistance of the bacteria to chlorhexidine action. In order to repair DNA lesions induced by disinfectants [34], *A. baumannii* uses the RecA protein for homologous recombination and recombination repair [4]. This strategy ensures bacterial survival as a nosocomial pathogen. *A. baumannii* also forms biofilm communities on most abiotic surfaces [35]. Bap_Ab_ has a role in cell–cell adhesion and is required for biofilm formation [32]. Biofilms increase *A. baumanni* tolerance to extracellular stress [35] and the action of antimicrobial agents [32].

So far, studies conducted on colistin-resistant mechanisms have not yet explained the mechanism of resistance associated with an increased tolerance of *A. baumannii* to the colistin action among strains circulating in South Africa [7,19]. To our knowledge, this study describes the first colistin-resistant *A. baumannii* isolate at the Doctor George Mukhari Academic Hospital (DGMAH) in Pretoria, South Africa, and investigates the use of adeABC efflux as a resistance mechanism, as well as determines the potential of the isolate to establish its niche within this tertiary health facility by evaluating its virulence factors.

## 2. Methods

### 2.1. Sample Collection

A tracheal aspirate was collected from a newborn male who was presented at the Neonate Intensive Care Unit at DGMAH and sent to the Doctor George Mukhari Tertiary Laboratory (DGMTL) for microbiology diagnostic testing. DGMTL is a level 3 clinical laboratory unit of the National Health Laboratory Services of South Africa, where routine laboratory diagnostics for patients received at DGMAH and surroundings clinics are performed. Ethical approval to conduct this research was granted by the Sefako Makgatho Health Sciences University Research Ethics Committee (SMUREC) with the following reference number, SMUREC/M/219/2020: PG.

### 2.2. Sample Identification and Antimicrobial Susceptibility Testing

The isolate was identified using a double identification method; VITEK 2 automated system (bioMerieux, Marcy-l’Étoile, France) and polymerase chain reaction (PCR) amplification of *bla_OXA-51_* gene [36,37,38]. Antimicrobial susceptibility testing was performed using VITEK 2 automated system (bioMerieux, Marcy-l’Étoile, France). Piperacillin/tazobactam, ceftazidime, cefepime, cefotaxime/ceftriaxone, imipenem, meropenem, gentamycin, ciprofloxacin, trimethoprim/sulfamethoxazole, tigecycline, and colistin were tested. Colistin resistance was confirmed using broth microdilution (ComASP^®^ Colistin 0.25–16 µg/mL, Diagnostic Liofilchem, Inc. Zona Industriale, Roseto degli Abruzzi, Italy), and performed and interpreted as described by the manufacturer.

### 2.3. Nucleic Acid Extraction

DNA and RNA extraction were performed as previously described by Nogbou et al., 2021 [29] following the boiling extraction method and RNA isolation Kit (ISOLATE II RNA Mini Kit, MagMAXTM Viral/Pathogen, bioline, London, UK), respectively.

### 2.4. Polymerase Chain Reaction for Molecular Detection of Oxacillinase (bla_oxa-51_), AdeABC Efflux Pump (adeB, adeR and adeS), and Plasmid-Mediated Colistin-Resistant Genes (mcr-1)

Gene amplification by conventional PCR was performed as previously described by Nogbou et al., 2021 [29]. The thermocycling conditions for conventional PCR and primer sequences used for detection of drug resistance are detailed in annexure 1.

### 2.5. Quantitative Real-Time PCR (qRT-PCR) Amplification of AdeABC Efflux Pump (adeB, adeR and adeS), and Plasmid-Mediated Colistin-Resistant Genes (mcr-1)

The qRT-PCR was conducted on cDNA as previously described by Nogbou et al., 2021 [29]. The thermocycling conditions and primers used are detailed in Appendix A, respectively.

### 2.6. Phenotypic Evaluation of AdeABC Efflux Pump adeB, adeS, and adeR Gene Expression

A functional AdeABC efflux system, used as a resistance mechanism, was assessed by evaluating the difference between the minimal inhibitory concentrations (MICs) for tigecycline (TGC) using the gradient diffusion method (tigecycline, MIC Test Strip, Liofilchem^®^ Srl, Roseto d’Abruzzi, Italy) before and after exposure to an efflux pump inhibitor, as described by Nogbou et al., 2021 [29].

### 2.7. Whole-Genome Sequencing

The purified genomic DNA for WGS was prepared using a combination of the boiling extraction method followed by DNA purification using the Quick-DNA™ Miniprep Plus Kit (Zymo-Spin™ Technology, ZYMO RESEARCH). The WGS was performed, as previously described by Mwangi et al., 2021 [39], at the Next-Generation Sequencing Unit at the University of the Free State.

### 2.8. Sequence Analysis and Typing

For WGS analysis and typing, the JEKESA pipeline (https://github.com/stanikae/jekesa accessed on 30 November 2021) was used. Briefly, Trim Galore v0.6.2 (https://github.com/FelixKrueger/TrimGalore accessed on 30 November 2021) was used to filter the sequence reads (Q, ≥ 20; length, ≥ 50), and de novo assembly was performed using SPAdes v3.13.2 (https://github.com/ablab/spades accessed on 30 November 2021); the assemblies were polished and/or optimized using Shovill v1.1.0 (https://github.com/tseemann/shovill accessed on 30 November 2021), and sequence typing was done using the multilocus sequence typing (MLST) tool v2.16.4 (https://github.com/tseemann/mlst accessed on 30 November 2021). Assembly metrics, including the GC content and number of contigs, were calculated using QUAST v5.0.2 (http://quast.sourceforge.net/quast accessed on 30 November 2021). All resultant contiguous sequences were annotated using the NCBI Prokaryotic Genome Annotation Pipeline v4.13 [40]. The antimicrobial resistance genes’ presence was corroborated using ABRicate (https://github.com/tseemann/abricate accessed on 30 December 2021) that included ARG-ANNOT [41], CARD [42], MEGARes [43], ResFinder [44], and AMRFinderPlus [45] databases. Virulence factor-associated genes were detected using Victors [46] and VFDB [47] databases.

### 2.9. Sequences and Genbank Accession Numbers

This whole-genome shotgun project has been deposited at DDBJ/ENA/GenBank under the accession JAKNTS000000000. The version described in this paper is version JAKNTS010000000 (Table 1).

A direct link to the deposit can be found at: https://www.ncbi.nlm.nih.gov/nuccore/JAKNTS000000000.1 (accessed on 13 March 2022).

## 3. Results

### 3.1. Isolate Identification and Antimicrobial Susceptibility Testing

The double identification method, using a VITEK2 automated system (bioMerieux, Craponne, France) and positive conventional PCR amplification of *bla_OXA-51_*, enabled us to identify the isolated strain as *Acinetobacter baumannii*. The strain taxonomic identity was confirmed using Kraken and BactInspectorMax following the whole-genome sequence analysis. The strain phenotypic antimicrobial susceptibility pattern showed resistance to all antibiotics tested with confirmed colistin resistance with a MIC greater than 16 µg/mL using broth microdilution.

### 3.2. Molecular Investigation of AdeABC Efflux Pump (adeB, adeR, and adeS) and Plasmid-Mediated Colistin-Resistant Genes (mcr-1)

The *adeB*, *adeR*, and *adeS* genes associated with an active efflux pump were detected using conventional PCR and qRT-PCR. The *mcr-1* plasmid-mediated colistin-resistant gene was not detected using conventional PCR and qRT-PCR.

### 3.3. Phenotypic Evaluation of AdeABC Efflux Pump adeB, adeS, and adeR Gene Expression

The assessment of a functional AdeABC efflux system as a resistance mechanism was conducted using tigecycline. The results revealed an MIC of 0.38 µg/mL before exposure to an efflux pump inhibitor and a MIC of 0.25 µg/mL after exposure to an efflux pump inhibitor.

### 3.4. Genomic Investigation of Resistance Mechanism

The whole-genome sequence was used to detect 19 additional antimicrobial resistance-associated genes (Table 2).

The whole-genome sequence detected 48 virulence factor-associated genes. The colistin resistance-associated gene, *mcr-1*, was not detected, as well as *lpxA*, *lpxC*, and *lpxD*. The *pmrB/pmrA/pmrC* operon-associated genes and genes associated with AceI protein production were also not detected.

### 3.5. Multi-Locus Sequence Typing

Following the Pasteur scheme, the isolated strain was identified as belonging to sequence type 2 (ST2), which is an ST belonging to clonal complex 2 (CC2).

## 4. Discussion

The rapid development of pan- and/or multi-drug resistance pattern among clinical isolates of *A. baumannii* is of concern worldwide [8]. The carbapenem-hydrolysing oxacillinase *bla_OXA-51_* gene has been reported to be intrinsic to *Acinetobacter* sp. and is recommended by several authors as a simple and reliable genomic identification feature of *A. baumannii* strains [36,37,38,48]. However, there is mention, within the literature, of *A. baumannii* strains not harbouring *bla_OXA-51_* [49]. In this report, the positive detection of the *bla_OXA-51_* gene was used as genomic confirmation for strain identification. This approach was consolidated by the whole-genome sequencing method that taxonomically confirmed strain identity, supporting recommendations made by previous researchers to use the *bla_OXA-51_* gene for *A. baumannii*’s identification [36,37,38]. 

An increase in incidences of colistin-resistant *A. baumannii* strains have been observed at South African tertiary health facilities [7,50]. The strain described in this study is of particular concern, as it showed resistance to all available drugs, including colistin and Tigecycline. Such strains pose a serious therapeutic challenge and the potential to cause devastating outbreaks [50,51]. The number and the diverse range of resistance-associated genes detected within its whole-genome analysis justified the phenotype results.

The targeted PCR and qRT-PCR amplifications of the *mcr-1* gene were negative. Moreover, no known colistin-associated resistance mutations were detected in the *lpx* or *pmr* genes. Furthermore, the whole-genome sequence analysis did not report any *mcr* genes. A similar observation was made by Snyman et al. [7] in Cape Town during their study conducted at the Tygerberg Academic Hospital while investigating 26 confirmed colistin-resistant *A. baumannii* isolates. Snyman et al. [7] supported that the absence of plasmid-mediated *mcr* genes and any known chromosomal mutations in *lpx* or *pmr* suggests that the colistin resistance in *A. baumannii* isolates may be due to a non-investigated mechanism. In agreement with this observation, Lean et al. [52] demonstrated that colistin resistance in *A. baumannii* is strongly associated with a change from histidine to tyrosine in position 181. However, the authors reported that six of fourteen confirmed colistin-resistant *A. baumannii* strains were not harboring this specific mutation [52]. This study revealed the presence of a mutation in the *lpsB* gene after whole-genome sequence analysis (Table 3 and Figure 1). These mutations are different from the mutation reported by Lean et al. [52]. Vijayakumar et al. [53] supported that the significance of the mutations in *lpsB* required more investigation to determine their implication in colistin resistance. Yet, the published data by these researchers support that colistin resistance in *A. baumannii* is more associated with molecular events within *lpsB* than *mcr* sequences [53]. Although further investigations are required for confirmation, the gathered evidence suggests that chromosomal mutations in *lpsB* might be responsible for colistin resistance in *A. baumannii*. *lpsB* enhances *A. baumannii’s* virulence in pulmonary infections [54,55]. The tracheal aspirate sample from which the strain described in this study was isolated supports the implication of *lpsB* in colistin resistance. This report suggests that the colistin-resistant strain within the South African tertiary hospital may be due to a mutation in the *lpsB* gene rather than the plasmid-mediated *mcr* genes’ acquisition or traditional *lpx* and *pmr* genes.

Other studies support that the active use of an efflux pump could be responsible for colistin resistance in *A. baumannii* [54,56]. This study reports a positive detection of AdeABC, AdeFGH, and AdeIJK efflux pump genes. Moreover, the result of the phenotypic evaluation of the AdeABC efflux pump, coupled with the mutations in the *adeR* and *adeS* genes, demonstrate an overexpression of the AdeABC efflux pump’s use as a mechanism of resistance to polymyxins. Studies have shown the use of an efflux pump as a resistance mechanism in bacteria improves bacteria survival in colistin stress [56,57,58,59]. 

The international *A. baumannii* clone II is associated with the production of OXA-23 carbapenem-hydrolysing oxacillinase [60] and is the predominant cause of outbreaks of *A. baumannii* infection [5]. The international clone II was identified as a high-risk clone, as it is one of the drivers of *A. baumannii’s* rapid dissemination across the world [61]. The isolated strain in this study was reported to belong to ST2, which is part of the international clone II and producer of *bla_OXA-23_* (Table 3). The presence of such a strain within a tertiary hospital is of serious concern, as the risk of an escalation in resistance to colistin among the species plausible, endangering patient life and bringing considerable risk within hospital environments.

The ability of *A. baumannii* to survive on inanimate objects and resist environmental stress enables the bacteria to colonize new environments and promote its success as a nosocomial pathogen [62,63]. An investigation of the virulence factors that enhance the strain’s survival in harsh environmental conditions was conducted to evaluate the extent of the threat that the colistin-resistant *A. baumannii* introduces to the health facility (Table 1 and Table 4). The investigated strain has *bap* and *recA* but not the *aceI* gene (Table 1 and Table 4). *A. baumannii* biofilm-associated protein production is mediated by the *bap* gene [32]. The protein enhances the development of high-order and complex *A. baumannii* communities’ structures on abiotic surfaces [32], such as catheters, endotracheal tubes, and other healthcare-associated equipment [32,64]. Biofilms provide a shielding effect to bacterial community members through the restriction of antimicrobial agents’ penetration [65]. The *recA* gene codes a DNA-damaged repair and recombination protein [4]. The gene is involved in SOS mutagenesis response and enhances bacteria survival against antimicrobial agents and oxidative stresses [4]. *A. baumannii* communities persisting within hospital environments become a source of infection to patients. Several other genes that are responsible for various virulence factors have been reported. The CsuA/BABCDE chaperone-usher pili assembly system, regulated by the BfmS/BfmR two-component system; the outer membrane protein, OmpA; the autoinducer synthase, AbaI, which is part of the quorum-sensing system and its repressor protein, AbaR; and the PgaABCD, which is responsible for the production of poly-β-1,6-N-acetylglucosamine, are all biofilm-related virulence factors responsible for the biofilm formation and cell–cell adhesion system [66]. Iron acquisition system-related genes have been identified within the genome of the study strain. Iron is a micronutrient essential for the growth of living organisms [67]. *A. baumannii*, like most aerobic bacteria, produces various high-affinity iron acquisition systems through the expression of the reported genes (Table 4). These systems will detect, trap, and present iron in a suitable form for bacterial use [68], enhancing the virulence and pathogenicity of the strain.

## 5. Conclusions

The genomic investigation of the first colistin-resistant *A. baumannii* isolated at this tertiary hospital in Pretoria revealed that the emergence of colistin resistance might be due to another resistance mechanism other than the widely reported *lpx* and *pmr* genes or the plasmid-mediated *mcr* genes. The diverse and multiple drug resistance-associated mechanisms expressed by the study strain, coupled with virulence factors that enhance its pathogenicity, survival in environmental stress, and niche establishment, indicate that this strain is a threat at this tertiary health facility.

## Figures and Tables

**Figure 1 antibiotics-11-00594-f001:**
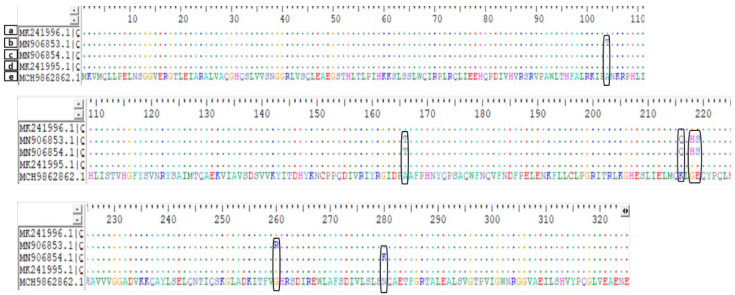
Mutations within *lpsB* sequence (e) compared with references of 4 sequences (a–d). The *lpsB* in our isolate is in contig 18 (JAKNTS010000018.1). Sequence (a–d) can be found in NCBI GenBank under their respective number. Mutations within the *lpsB* gene have been associated with resistance to colistin, with sequences showing any mutations within this gene most likely to be resistant to colistin.

**Table 1 antibiotics-11-00594-t001:** Specification of investigated strain.

Organism	*Acinetobacter baumannii*
Strain	SMU.6245.Ab.ND.2021
Sequencer	Illumina MiSeq
Data format	Assembled
Experimental Factors	Genome sequence of pure microbial culture
Experimental Features	Genome sequence followed by assembly annotation
Consent	N/A
Sample source	tracheal aspirate Homo sapiens

**Table 2 antibiotics-11-00594-t002:** Genome characteristics and resources.

N	Name	Genome Characteristics and Resources
1	NCBI BioProject	PRJNA803988
2	NCBI BioSample ID	SAMN25694890
3	NCBI genome accession Number	JAKNTS010000000
4	Sequences type	genome
5	Total number of reads	1,280,666
6	Clean reads	1,260,538
7	Overall coverage	74.0×
8	Estimate genome size	4,025,130
9	G + C content (%)	38.84
10	Genes (total)	3906
11	tRNAs	62
12	rRNAs	1, 2 (16S, 23S)
13	ncRNAs	4
14	Pseudo Genes (total)	63

**Table 3 antibiotics-11-00594-t003:** Additional antimicrobial resistance-associated genes investigated using whole-genome sequencing.

Resistance	Acquired Resitance Genes
Aminoglycoside	*aph(6)-Id*; *aph(3″)-Ib*; *ant(3)-IIa* and *armA*
Beta-lactam	*bla_OXA-66_*; *_blaOXA-23_*; *bla_ADC-25_*; *bla_ADC-73_*; *bla_A1_*; *bla_A2_* and *bla_MBL_*
Fosfomycin	*abaF*
Macrolide	*msrE* and *mphE*
Polymixin	*lpsB*
Streptogramin	*strA* and *strB*
Sulphonamide	*sul2*
Tetracycline	*tetB*

**Table 4 antibiotics-11-00594-t004:** Virulence-associated genes investigated using whole-genome sequencing.

Virulence Factors	Virulence-Associated Genes
Biofilm formation system, cell–cell adhesion	*bap*, *bfmR*, *bfmS*, *csuA*, *csuA/B*, *csuB*, *csuC*, *csuD*, *csuE*, *pgaA*, *pgaB*, *pgaC* and *pgaD*
Quorum sensing	*abaI* and *abaR*
Resistance-nodulation-division AdeFGH and AdeABC efflux pump	*adeF*, *adeG, adeH* and *adeL*; *adeB*, *adeS*, and *adeR*
Resistance-nodulation-division AdeIJK	*adeI*, *adeJ*, *adeK, adeN*
Multi-drug and toxic compound extrusion	*AbeM*
Small multi-drug resistance transporters	*AbeS*
Iron acquisition systems	*barA*, *barB*, *basA*, *basB*, *basC*, *basD*, *basF*, *basG*, *basH*, *basI*, *basJ*, *bauA*, *bauB*, *bauC*, *bauD*, *bauE*, *bauF* and *entE*
Phospholipase	*plc*, *plcD*
Porin	*OmpA*
DNA recombination	*recA*
Regulator of the MexEF-oprN efflux pump in *Pseudomonas aeruginosa*	*mexT*

## Data Availability

Data is contained within the article.

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
