# Peer review of "Whole-Genome Sequencing of a Colistin-Resistant Acinetobacter baumannii Strain Isolated at a Tertiary Health Facility in Pretoria, South Africa"

_antibiotics, 2022, doi:10.3390/antibiotics11050594_

Round 1

Reviewer 1 Report

Abstract:

Results section needs to be revised keeping in view conclusion. Statement of conclusion is not justified based on results. Moreover grammatical typos exist in abstract.

Keywords needs to be revised.

Background:

This section is unnecessarily too large. Concised description should be observed with no more than one and half page (3,4 paragraphs)

Materials and Methods;

It is lacking information about number of samples collected, number of samples proceeded for molecular analysis.

Moreover description of methods are unnecessarily expanded. These need to be concised keeping in view necessity of protocol.

Discussion:

Discussion section is too long for any research paper. Discussion should be revised keeping in view objectives of the study/results of the study.

Author Response

Thank you for your comments. See attached document for responses. 

Reviewer 2 Report

The authors are highly appreciated for the nice study. However, they are suggested to take care of the followings-

1. Introduction did not reflect the overall merit of this research

2. The result is very poorly described. 

3. The genomic information especially the Accession number (bioproject/ biosample) was not provided. Thus, genomic information provided in the manuscript couldn't be evaluated.

4. The authors have suggested/assumed an alternative mechanism of colistin resistance through lpsB. They have mentioned specific mutation in the lpsB gene probably responsible for colistin resistance, however, they didn't provide any evidence or data such as alignment data showing mutation in support of their statement!!

Author Response

(The authors gave the same response as above.)

Reviewer 3 Report

File attached herewith.

Author Response

(The authors gave the same response as above.)

Reviewer 4 Report

This manuscript describes the mechanism of colistin-resistant Acinetobacter baumannii isolated at a tertiary hospital in South Africa. Colistin-resistant A. baumannii was isolated from a new-born male's tracheal aspirate. Piperacillin + Tazobactam, Ceftazidime, Cefepime, Cefotaxime/Ceftriaxone, Imipenem, Meropenem, Gentamycin, Ciprofloxacin, Trimethoprim/Sulfamethoxazole, Tigecycline and Colistin were all resistant to this strain. adeB, adeR and adeS were found, while mcr-1 was not detected. A whole genome sequence was done to reveal the details of antibiotic resistance and virulence associated genes. And this strain belongs to ST2.

The manuscript was well written but did not show some results. So, I have a few comments that give the reader a clear message.

1. The authors should provide the AST results, particularly the MIC of colistin.

2. Whether the AdeABC efflux pump is expressed in high or low level should be shown.

3. line 235, recheck 21 or 19 antimicrobial resistance genes.

4. The authors should submit the genome sequence data to GenBank.

Author Response

(The authors gave the same response as above.)

Round 2

Reviewer 1 Report

Thank you for improving manuscript. 

Author Response

Thank you for the valuable comments that have improved the manuscript.

Reviewer 2 Report

  • The authors are highly appreciated for the improvement made throughout the Manuscript. However, the introduction section is still poor and feels distorted as you can see from their writing (highlighted) below-

So far studies conducted on Colistin resistance mechanisms have not yet explained the mechanism of resistance associated with increased tolerance of A. baumannii to Colistin action among strains circulating in South Africa. To our knowledge, this study describes the first Colistin-resistant A. baumannii isolate at Doctor George Mukhari Academic Hospital (DGMAH) in Pretoria, South Africa, investigates the use of adeABC efflux as a resistance mechanism and determines the potential of the isolate to establish its niche within this tertiary health facility by evaluating its virulence factors. 

Is adeABC efflux pump responsible for increased tolerance of A. baumanii to Colistin?

  • In the methodology section 2.6 adeABC pump phenotypic expression was determined against  tigecycline!! If they want to prove adeABC efflux pump's association with increased tolerance to Colistin, then, why didn't they assay MIC of Colistin before and after exposure to an efflux pump inhibitor?? It's very confusing to me.

In addition,

  1. No need to provide link of the GenBank deposition in different section. The link could be simply provided in parenthesis at the end to section 2.9.
  2. No need to provide Table 1.
  3. Position of Table 2 is very illogical. Perhaps the information are not that critical for this manuscript.
  4. Result section is yet very poorly described. I would rather suggest the authors to follow manuscripts describing such data in other studies.
  5. Legend in the Figure 1 is not sufficient. There is no indication regarding the specific substitution which is responsible for colistin resistance. The authors should indicate which of the a, b, c or d are colistin resistant and which are sensitive. From the alignment it is evident that sequence "a" and "d" are identical to the study sequence "e". As the study strain showed colistin resistance, my understanding is "a" and "d" are also colistin resistant and the other two are not. Am I right? If not, please provide enough information to make it logically understandable. 
  6. In section 3.3 the author did not mention the antibiotic, i.e. MIC against which antibiotic? From materials and method the antibiotic should be Tigecycline.
  7. In line no. 393 to 400 the authors described an over expression of AdeABC efflux system and claimed its association with resistance to polymyxin. Is it not illogical? May be it is right, however, the authors did not show any data in support of this statement. 

Overall,  I am suggesting the authors to take time and make logical modifications as they have wonderful data to be published!!

Author Response

Manuscript Antibiotics-1658602

Reviewer2

Thank you for your valuable comments. Below are our responses.

Reviewer comment: Is adeABC efflux pump responsible for increased tolerance of A. baumanii to Colistin?

Author response: The efflux pumps are a universal mechanism of resistance that affect all classes of drugs indiscriminately. In investigating this strain, it was shown phenotypically that the strain uses this mechanism of resistance.

Reviewer comment: In the methodology section 2.6 adeABC pump phenotypic expression was determined against  tigecycline!! If they want to prove adeABC efflux pump's association with increased tolerance to Colistin, then, why didn't they assay MIC of Colistin before and after exposure to an efflux pump inhibitor?? It's very confusing to me.

Author response: Both tigecycline and colistin are polymyxins. There mode of action is thus the same and the mechanism of resistance against them especially in relation to activity of efflux pumps is thus the same. For the purpose of investigation, colistin resistance is recommended to be performed using broth microdilution; this methodology is much more difficult to demonstrate the use of efflux pumps as opposed to a methodology employing an E-Strip that is available for tigecycline as was done in this study.

Reviewer comment: No need to provide link of the GenBank deposition in different section. The link could be simply provided in parenthesis at the end to section 2.9.

Author response: The link and details to the sequence as deposited in GenBank is now only under section 2.9.

Reviewer comment: Position of Table 2 is very illogical. Perhaps the information are not that critical for this manuscript.

Author comment: The table summaries the details of the whole sequence as requested by other reviewers. We thus have left it as is.

Reviewer comment: Result section is yet very poorly described. I would rather suggest the authors to follow manuscripts describing such data in other studies.

Author comment: We appreciate the suggestion. We have laid out the results in reference to similar previously published work.

Reviewer comment: Legend in the Figure 1 is not sufficient. There is no indication regarding the specific substitution which is responsible for colistin resistance. The authors should indicate which of the a, b, c or d are colistin resistant and which are sensitive. From the alignment it is evident that sequence "a" and "d" are identical to the study sequence "e". As the study strain showed colistin resistance, my understanding is "a" and "d" are also colistin resistant and the other two are not. Am I right? If not, please provide enough information to make it logically understandable. 

Author response: Strains with mutations within the LpsB gene have been shown to have a greater predisposition to colistin resistance. Without particularly referring to a single mutation being responsible to this outcome. Previous published investigations indicate greater involvement of mutations within this gene to colistin resistance as alluded to in the discussion section.

Reviewer comment: In section 3.3 the author did not mention the antibiotic, i.e. MIC against which antibiotic? From materials and method the antibiotic should be Tigecycline.

Author comment: The antibiotic used for the investigation has now been indicated as tigecycline in section 3.3 as advised.

Reviewer comment: In line no. 393 to 400 the authors described an over expression of AdeABC efflux system and claimed its association with resistance to polymyxin. Is it not illogical? May be it is right, however, the authors did not show any data in support of this statement.

Author comment: over expression of AdeABC efflux pump system is a well described mechanism of resistance employed by may bacteria to indiscriminately resist the action of different classes of antibiotics. In this manuscript we investigated its use against tigecycline a polymyxin and indicated the outcome in section 3.3.  

We once again appreciated the comments contributing to a better manuscript.

Reviewer 3 Report

All the changes made are accepted.

Author Response

(The authors gave the same response as above.)

Reviewer 4 Report

I appreciate the author considering my previous suggestion. However, I have some additional comments.

  1. The background is too long. A maximum of 400-500 words is enough.
  2. Section Methods; 2.10 and Table 1, I would suggest to leave it to supplement.
  3. Please recheck 48 or 51 virulence factor associated genes.

Author Response

Manuscript Antibiotics-1658602

Reviewer4

Reviewer comment: The background is too long. A maximum of 400-500 words is enough.

Author response: Thank you for highlighting this. The section has been significantly reduced to keep information that is enough to introduce the subject of the manuscript.

Reviewer comment: Section Methods; 2.10 and Table 1, I would suggest to leave it to supplement.

Author response: Section 2.10 has been cancelled as advised. The data and table 1 were requested by other reviewers. We have thus left it as is however we have no reservations for this table to be presented as a supplement in the final publication. We therefore leave this discretion to the editor.

Thank you again for the valuable comments.

This manuscript is a resubmission of an earlier submission. The following is a list of the peer review reports and author responses from that submission.